# Advances in Material Nanosensitization: Refractive Property Changes as the Main Parameter to Indicate Organic Material Physical–Chemical Feature Improvements

**DOI:** 10.3390/ma15062153

**Published:** 2022-03-15

**Authors:** Natalia V. Kamanina

**Affiliations:** 1Lab for Photophysics of Media with Nanoobjects, Vavilov State Optical Institute, Kadetskaya Liniya V.O., Dom 5, Korp.2/Babushkina Str., Dom 36, Korp.1, 199053 St. Petersburg, Russia; nvkamanina@mail.ru; 2Photonics Department, Electronic Faculty, St. Petersburg Electrotechnical University, (“LETI”), Ul. Prof. Popova, Dom 5, 197376 St. Petersburg, Russia; 3Branch of National Research Center «Kurchatov Institute», Petersburg Nuclear Physics Institute, 1 md. Orlova Roshcha, 188300 Gatchina, Russia

**Keywords:** functional nanomaterials, organics, sensitization, refractive index change, polyimide, liquid crystal, fullerenes, carbon nanotubes, reduced graphene oxides, surfaces

## Abstract

In the current paper, the results of the sensitization process’ influence on the refractive organic materials’ features are shown. The correlation between the refractive properties and the intermolecular charge transfer effect of doped organic thin films are shown via estimation of the laser-induced change in the refractive index. The refractive parameter is shown for a model organics matrix based on a polyimide doped with fullerenes, carbon nanotubes, reduced graphene oxides, etc. A second harmonic of the Nd-laser was used to record the holographic gratings in the Raman–Nath diffraction conditions at different spatial frequencies. The laser-induced refractive index change was considered to be an indicator in order to estimate the basic organic materials’ physical–chemical characteristics. Additional data are presented for the liquid crystal cells doped with nanoparticles. The correlation between the content of the nanoobjects in the organics’ bodies and the contact angle at the thin film surfaces is shown. Some propose to use this effect for general optoelectronics, for the optical limiting process, and for display application.

## 1. Introduction

It is well known that in the past 10–15 years, the influence of nanostructuration on the basic physical and chemical properties of materials, especially on the features of organic ones, is equipped with unique assets [1,2,3,4,5,6,7,8,9,10,11,12,13,14,15,16,17]. Moreover, the modification of the interface with the nanoparticles is also taken into account when researchers discuss the nanocomposites’ properties change [18,19,20]. As according to the local opinion of the author of this paper, it is connected with the important dependences between the refractive parameters and the spectral, photoconductive, dynamic, etc., characteristics [21,22,23]. This vision can be illustrated in Figure 1.

This dependence can be more efficiently shown in the conjugated organics with an initial donor–acceptor interaction. The intermolecular charge–transfer complex (CTC) formation can be formed via an introduction in the organic matrix with nanoobjects such as fullerenes, carbon nanotubes (CNTs), reduced graphene oxides (RGOs), shungites (Shs), and quantum dots (QDs). A possible high-quality model is shown in Figure 2. It should be remarked that a simple scheme between an interaction with the donor part of the matrix molecules and the fullerenes, as the intermolecular acceptors, via a change in the refractive index has previously been shown in [23]. For example, if one uses polyimide conjugated structures, the monomeric links of polyimides are intramolecular donor–acceptor (D–A) complexes with the charge transfer between the donor and acceptor molecular fragments, which can be changed. Polyimides consist of the acceptor diimide fragments with an electron affinity energy of 1.12–1.46 eV and of the donor fragments: triphenylamine (TPA), carbazole, fluorene, and benzene with an ionization potential of 6.5–9.2 eV. A Low ionization potential allows for D–A complexes to be formed between the monomeric links of the polyimide and incorporated into acceptor molecules during both structural–chemical and injection sensitizations. According to this model [23], in the donor–acceptor conjugated materials, the electrons will be captured predominantly by the introduced acceptor (for example, fullerene) with a higher electron affinity energy than that for the intramolecular acceptor in the matrix organic structure. That is, a new charge transfer path with a long electron free pathway and a larger charge will be formed, since, say, a fullerene, as an intermolecular acceptor, can capture not one but six electrons. This intermolecular CTC dramatically influences the nonlinear optical, dynamic, and the photoconducting properties of the materials including the liquid crystals as well, which have been actively discussed. It should be taken into account that the nano- and bioparticles can be considered as effective dopants of the organics systems.

To study the nanostructured materials and to reveal the intermolecular CTC formation and its influence on the refractive characteristics, the different techniques and approaches can be used. Among them, namely, the laser technique (i.e., the Z-scanning scheme, third harmonic generation, and four-wave mixing technique can be used) has been successfully applied due to the reason that the lasers are operated at different spectral ranges and at different energy densities (i.e., power). Thus, it can provoke the activation of the absorption not only at the edge of the absorption band of the model matrix materials but in the IR spectral range due to the bathochromic spectral shift in the nanostructured compounds [24]. Moreover, it can reveal the dramatic change in the laser-induced refractive index and other nonlinear parameters, for example, such as the nonlinear refraction coefficient and the cubic nonlinearity [25,26].

In this aspect, it should be remarked that to analyze the nonlinear optical processes one should take into account that when the electric field of the laser wave is less than an intra-atomic electric field correlated with an electron charge and with the Bohr radius, we should estimate the linear effect. But, when the electric field of the laser wave is larger than an intra-atomic electric field, we should draw the attention on the nonlinear optical features. Using this aspect, the values of the optical susceptibility play an important role in the nonlinear optical effect. Really, the most important optical characteristic of all inorganic or organic materials with different symmetry is the induced dipole, which can be expressed through the dipole polarizabilities α^(n)^. These are, in turn, related by the proportional dependence to the nonlinear susceptibility, χ^(n)^, and to the local volume, υ, of the materials (media). Thus, the laser–matter interaction provokes the change in the polarization of the media and predicts the change in the important properties such as the photorefractive, dynamic, and photoconductive ones. The change in the photorefractive properties can be easy visualized via the holographic recording technique used [27].

The changes in the diffraction efficiency and the coefficients of the nonlinear refraction and the cubic susceptibility have been observed in the systems based on the poly(vinyl carbazole), polyimide (PI), pyridines, polyaniline, and other polymer matrices as well as in the liquid–crystalline materials sensitized by the abovementioned nanoobjects [28,29,30]. The investigations performed by the author with coworkers have also contributed to the research in this field of materials science.

There are some discrepancies in the published data on photorefraction, which are related to the difficulties encountered in the attempts to directly separate the mechanisms of diffusion and the drift of the charge carriers. Moreover, it is not easy to strictly determine the electron path lengths in the charge transfer process in the systems with the initial donor–acceptor interaction under the conditions in which an intermolecular complex formation is dominated over the intramolecular processes. These discrepancies hinder an adequate explanation of the differences in the nonlinear optical parameters observed at the various spatial frequencies for the same conjugated matrix and at the same concentration of the sensitizer.

This paper presents the analysis of the results and the systematization of the data on the photorefraction in the application to the PI–fullerenes, PI–carbon nanotube, and LC–NP-doped model system. Some other nanoparticles were used for comparison. On this basis, the possible influence of the arrangement of a sensitizing nanoobject and the charge-transfer pathway during the intermolecular complex formation on the nonlinear optical properties, namely, on the laser-induced refractive index of the organic nanocomposites was considered. Moreover, the change in the established tendency of the materials’ relief due to the change in the nanoparticles’ concentration can additionally be shown.

## 2. Materials and Methods

The basic photorefractive characteristics were studied using the four-wave mixing technique shown previously in [31]. The second harmonic of the nanosecond pulsed Nd-laser at a wavelength of 532 nm was used. The pulse durations were within of τ ~ 10–20 ns. The energy density was chosen in the range of 0.1–0.7 J·cm^−2^. The amplitude-phase thin gratings were recorded under the Raman–Nath diffraction conditions at the spatial frequency Λ of 90, 100, 150, and 170 mm^−1^ according to which Λ^−1^ ≥ *d*, where Λ^−1^ is the inverse spatial frequency of recording (i.e., the period of the recorded grating), and *d* is the organic film thickness. It should be remembered that the photorefractive parameters were revealed not only for the nanosensitized conjugated films but for the pure ones too for comparison. It should be taken into account that this laser technique has some advantages in comparison with the third harmonic generation scheme due to the fact that the first technique permits the testing of materials in the reversible mode without their distraction. Some view of the setup used in the current experiments is shown in Figure 3. An Nd-laser (1) with a passive Q-factor modulator operating at a wavelength of 532 nm was used to generate recording beams. A plane-parallel glass plate (2) branched off part of the flow to the synchronization unit (9). Mirrors (3) formed and directed the recording beams to the sample under study (7). The diaphragms (4) blocked the spurious reflections from the mirrors (3). The diaphragm (8) absorbed the recording beams that passed through the sample. A continuous diode-pumped neodymium laser (10) operating at the second harmonic was used for the reading. The reading beam diffracted on the recorded lattice, and the beams corresponding to the zero and first order of diffraction using mirrors (5) were diverted to the photo sensors (6). The scheme allowed for a variation of the spatial frequency of the recorded grating by moving the mandrel with the sample relative to the forming mirrors (3) with subsequent adjustment of the installation.

The experimental investigations were performed on 2–5 μm thick thin PI films containing triphenylamine and the diimide fragments that played the roles of an intramolecular donor and an acceptor, respectively. The PI matrix was doped by various dispersed sensitizing additives based on nanoobjects such as the fullerenes, QDs, shungites, CNTs, and RGOs. Nanosensitizing additives of the latter types were advantageous in being inexpensive, as they were available from domestic manufacturers and ensured good reproducibility of the experimental data (comparable in this respect to the results obtained with the pure fullerenes and the single-walled nanotubes). The concentrations (*c*) of fullerenes, QDs, shungites, CNTs, and RGOs amounted to 0.2–0.5; 0.03–0.003; 0.1–0.2; 0.05–0.1; 0.1 wt.%, respectively (relative to the dry matrix substance). The thin polyimide films were prepared using tetrachloroethane as a solvent. It should be remarked that this solvent was a good matrix solution for the different polymers and the fullerenes as well [32]. It should be noticed that the polyimide materials are studied by different scientific and technical teams in the word [33,34,35,36,37,38] due to the fact that this polymer matrix has unique high melting point [38] at more than 900 degrees; thus, it can be treated by different methods and techniques with good advantages. Polymer-dispersed liquid crystal (PDLC) cells were constructed in an *S*-configuration with a thickness of 10 microns. The content of the nanoparticles in the PDLC matrix varied from 0.1 to 5 wt.%.

## 3. Results and Discussion

The basic results of this study are shown in Table 1 [39,40,41,42,43,44,45,46]. The data presented are summarized according to the comparative data on the photoinduced changes of the refractive index, Δ*n*_i_, for the model conjugated PI-based structure sensitized by the various nanoobjects.

Let us pay an attention to the fact that the change in the refractive index depended on the concentration of the sensitizer and on its nature as well as on the spatial frequency at which the information was recorded. Moreover, the change in the spatial frequency was connected with the change in the dominant role of the diffusion and/or drift mechanism of the charge spreading. Really, in the case of the nanocomposite irradiated at small spatial frequencies (the large periods of recorded grating), the drift mechanism of the carrier spreading in the electric field of an intense radiation field will be most likely predominant, while at the large spatial frequencies (the short periods of the recorded grating) the dominating process was the diffusion. Furthermore, it is natural to suggest that the variations in the angle of the nanoobject orientation relative to the intramolecular donor can probably significantly change the pathway of the charge carrier transfer, which will lead to changes in the electric field gradient, dipole moment (proportional to the product of the charge and the distance), and the charge carriers’ mobility as well. In order to more clearly explain this evidence, the various possible options for the location of a molecule or a charged particle near the top of the CNTs are shown in Figure 4.

It should be remarked once again that the considered polyimide systems are the materials with an initial intra-molecular CTC formation process. The main link of the studied polyimide molecules contains a triphenylamine fragment as a donor and a diimide fragment as an acceptor. The initial acceptor fragment of the polyimide molecules had an electron affinity energy close to 1.12–1.46 eV [47], which was approximately two times less than the one for the fullerenes [48], which released an electron affinity of 2.6–2.7 eV for C_60_ and 2.8 for C_70_, respectively. Under the sensitization process, the charge transfer revealed from the intra-molecular donor fragment of the organic conjugated molecules not to its acceptor fragment but to the nanoobjects if the electron affinity energy of the nanoobjects was larger than the one for the intra-molecular acceptor fragment. Moreover, as has been shown in [49], fullerenes, for example, can accept not one but six electrons. Thus, via an intermolecular CTC, the creation of a larger dipole moment can be possible. This dipole moment, µ_inter_, based on the intermolecular CTC is essentially larger than the one (µ_intra_) obtained from the intra-molecular process and can be increased up to one order of magnitude. The first supporting experimental result devoted to the measurement of the dipole moment in the pure and in the sensitized composite based on the polyimide has been obtained previously and shown in [47], supported in paper [50], and presented by the model above shown in Figure 2.

Let us support the important role of the refractive index change to the features of the structured organics materials with the intermolecular charge transfer complex formation process by adding data on other conjugated structures such as 2-cyclooctylamino-5-nitropyridine (COANP), N-(4-nitrophenyl)-(L)-prolinol (NPP), 2-(n-prolinol)-5-nitro-pyridine (PNP), and polyaniline (PANI) [51,52,53,54]. These structures have been studied for optical restriction (limiting), for use in the optically and electrically addressed light modulators, and for amplitude-phase holograms recording in the different spectral and energy-density ranges. For all materials, the dependence of the change in the refractive index during doping by the nanoparticles with the changes in the nature and concentration of the nanoparticles, the spatial frequency of the recording, and the levels of the incident energy density have been found.

To summarize the influence of the intermolecular CTC formation on the basic properties of the sensitized organics, it should be testified that the data presented in the Table 1 can successfully support the influence of the intermolecular CTC on the change of the laser-induced refractive index, Δ*n*_i_, that correlated with the local volume polarizability changing. Actually, to estimate Δ*n*_i_, the diffraction efficiency, η, in the first diffraction order was measured. The mathematical procedure to determine the relation among these values is shown in [55] and applied for the nanostructured matrix in [23,39,40] via Equation (1).
(1)$$\eta =\frac{I_{1}}{I_{0}}={\left(\frac{\pi \Delta n_{i}d}{2\lambda }\right)}^{2}$$
where Δ*n*_i_ is the induced change in the refractive index, *I*_1_ is the intensity in the first diffraction order, *I*_0_ is the input laser intensity, *d* is the thickness of the medium, and λ is the wavelength of the light incident on the medium. After that, the nonlinear refraction coefficient, *n*_2_, and cubic nonlinearity, χ^(3)^, can be calculated via Equations (2) and (3):(2)$$n_{2}=\frac{\Delta n_{i}}{I}$$
(3)$$\chi^{\left(3\right)}=\frac{n_{2}n_{0}c}{16\pi^{2}}$$
where *n*_0_ is the linear refractive index, *n*_2_ is the nonlinear refractive index, *c* is the light velocity, and χ^(3)^ is the cubic nonlinearity.

It should be analyzed and mentioned that the nonlinear optical coefficients for the nanoobject doped organics of the current study were close to each other and can be found in the following range: *n*_2_ = 10^−10^–10^−9^ cm^2^·W^−1^ and χ^(3)^ = 10^−10^–10^−9^ cm^3^·erg^−1^. Moreover, it is important to note that these values can be compared with the ones for the inorganic volumetric nonlinear optical crystal, for example, for LiNbO_3_ and for the Si-based materials. For example, in [55], the nonlinear parameters of the inorganic Si materials are shown; these data visualized the following characteristics: *n*_2_ ~ 10^−10^ cm^2^·W^−1^ and χ^(3)^ = 10^−8^ cm^3^·erg^−1^. Moreover, the data for the often used SiO_2_ systems are presented as well; they are: *n*_2_ = 10^−16^ cm^2^·W^−1^ and χ^(3)^ = 10^−14^ cm^3^·erg^−1^. However, it should be taken into account that the inorganic structures have dimensions close to several millimeters or tens of micrometers, while the sensitized organics films have dimensions in the range of 2–10 μm, which is less dramatically. Thus, it can provoke to place these nonlinear optical organic films in the complicated specific laser schemes or in the general optoelectronics links.

Furthermore, some uses for non-toxic bio-objects in optoelectronics are shown in [56]. PDLC doped with DNA has shown change in refractive coefficients of 1.39 × 10^−3^; that is, it coincides with the values obtained for the nanoparticle-doped PDLC (see Table 1). Here, the significant advantage of biosensitization is its non-toxicity; however, there is a problem in the conservation of the biological objects in order to preserve their properties.

Some consideration about the possible destruction of the polyimide materials should be taken into account. This unique material, apparently the best of polymer compositions, has a melting point of more than 900–1000 degrees. In Kamanina’s team, these materials poured onto the substrates have been studied both recently and 5–15 years ago. When structuring with carbon nanoparticles, the material becomes more durable and does not lose its properties after exposure to laser radiation at the wavelength of 532 nm. As for the pure polyimide composition, it should be mentioned that earlier, the destruction limit was found (during the transition from the reversible to the irreversible recording mode of the amplitude-phase hologram). This limit is 0.5–0.6 J·cm^−2^ for the irradiation at a wavelength of 532 nm [57].

Recently, we started the irradiation of polyimide in the vacuum UV range [58] in order to propose this procedure for the liquid crystal orientation. In this range, we found some destruction of the polyimide materials at the wavelength of 126 and 172 nm due to the fact of some ablation process. We now collect data regarding irradiation of polyimide, COANP, NPP, PNP, etc., materials and will publish a paper in the future according to our consideration about the features of the abovementioned materials treated under the UV conditions.

It should be mentioned that an introduction to the nanoparticles, for example, fullerene, can predict the change not only in the basic properties of the body of the materials but can reveal the modification of their surfaces. Firstly, this effect has been predicted and shown in [59] in which it was established that the doping process efficiently influences the surface relief of the polyimide materials. The wetting (contact) angle was measured in order to use it as an indicator of the relief roughness change. The measurement of the wetting angle at the organic solid thin polyimide film surface clearly visualized this fact. Actually, the wetting angle at the surface of the doped polyimide thin films changed from 72° to 73° (pure polyimide material); to 86–87° (at 0.1 wt.% C_70_ additives in PI); to 89–90° (at 0.5 wt.% C_70_ additives in PI); up to 102–103° when the content of the fullerene C_70_ in the polyimide was 1.0 wt.%. Thus, despite the good uniformity of the doped organic polyimide film, the fullerene skeleton can effectively influence the surface. This effect has been recommended to be used to orient the liquid crystal (LC) materials. This idea, namely, the correlation between the variation of the content of the nanoparticles in the matrix body and the surface roughness change is connected with the variation in the orientation of the LC dipoles, has been additionally discussed in [60] and extended for the optical limiting use in [61] to propose the novel additional optical limiting mechanism, when the energy losses via reflection from the modified surfaces (skeleton of the nanoparticles is effectively appeared) can be taken into account under the condition of the material body sensitization with the varied content of the nanoobjects. It should be mentioned that the surface relief change visualization can be efficiently used in the education process due to the fact that these experiments are not so complicated, but the results are adequate and can be obtain easy with good advantage. Furthermore, it should be noticed that the relief roughness change should be studied in more detail and the oxidation effect, free volume, skin layer changes, etc., characteristics taken into account, but these can be considered in the future papers.

## 4. Conclusions

Thus, the first important fact based on the analysis of the obtained results, leads to the following conclusions. Taking into account the experiments on changing the refractive index, not only in the polyimide matrix but also when testing other conjugated materials that have the *intra*molecular donor–acceptor interaction and allow its change to the *inter*molecular one when doped with the nanoparticles, it is important to note and strengthen the key role of this parameter, namely, the refractive index. The refractive index change can really be considered as an indicator of other organic physical–chemical parameters changes. The doping process of the organics with the shown nanoobjects significantly influences the photorefractive properties of the matrix materials, for example, based on PI, COANP, liquid crystals, etc. An increase in the electron affinity (shungite, fullerenes, QDs, etc.) and specific area (CNTs, RGQ, etc.) implies a dominant role of the intermolecular processes leading to an increase in the dipole moment, local polarizability (per unit volume) of the medium and the mobility of the charge carriers as well. A change in the distance between an intramolecular donor and intermolecular acceptor as a result of variation in the arrangement (rotation) of the introduced nanosensitizer leads to changes in the charge transfer pathway in the nanocomposite. Varying the spatial frequencies permits the realization of the different mechanisms for the charge carriers moving in the doped organics. Different values of the nonlinear optical characteristics in the systems with the same sensitizer type and the same concentration can support competition between the carrier drift mechanism and the diffusion processes in the organic nanostructures under the action of the laser radiation. The special role of the dipole moment as a macroscopic parameter of a medium accounts for a relationship between the photorefraction and photoconductivity characteristics, which suggests expansion of the application area of nanoobject-doped organics in general optoelectronics and solar energy devices. It has been shown the priority use, namely, the doped organics materials in comparison with the often applied inorganic Si or SiO_2_ structures.

It should be taken into account once again that for all materials considered, as well as for the other organics with the *inter*molecular charge-transfer process realized via the sensitization with the NPs, the dependence of the change in the refractive index during doping by the nanoparticles with the changes in the nature and concentration of the nanoparticles, the spatial frequency of the recording, and the levels of the incident energy density should be found.

The second important fact is based on the specific sensitization process influence on the modification of the organic thin film relief. It can be found via the NPs’ skeleton visualization at the organic films’ surfaces. This effect can be useful for the laser physics to attenuate the laser beam to protect the human eyes and technical devices from high laser irradiation and can be applied in the display technique in order to use the novel relief for the LC molecules’ orientation. The variation in the content of the nanoparticles in the materials body can vary the wetting angle at the films’ surfaces. Thus, orthogonal (homeotropic) orientation can be established, and the new relief can be considered as the alternative approach for the MWVA display technique. Moreover, due to the fact that the surface modification can be easy visualized, it is useful in the education process to show students the operation of the liquid crystal cells.

Finally, it should be mentioned that this paper included the results of the analysis of many experiments and calculations carried out with different organic structures when doping them with the different nanoparticles with varied concentrations and under the different exposure conditions. That is, here, the author tried to find an approach to assessing the effectiveness of the sensitization of the conjugated organic systems by the nanoparticles with an emphasis on changing the refractive index as a basic material science parameter.

## Figures and Tables

**Figure 1 materials-15-02153-f001:**
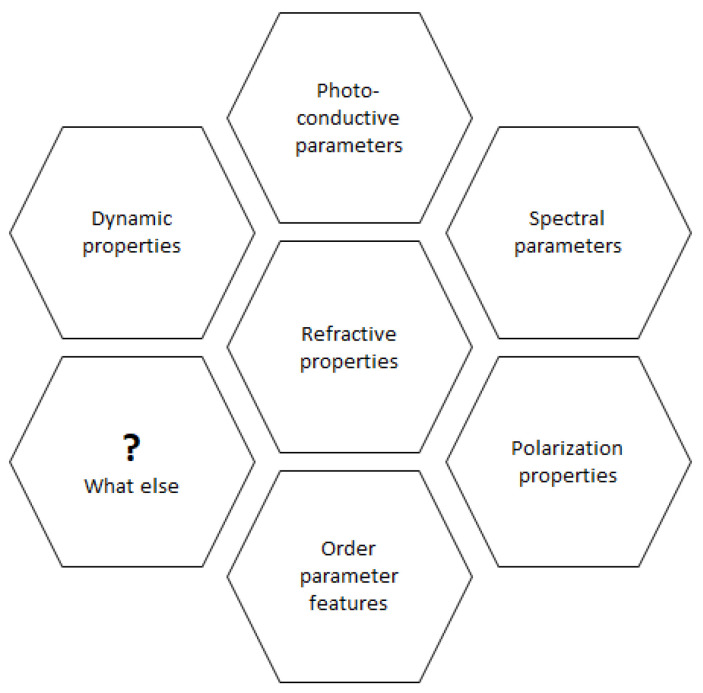
The qualitative (model) representation of the relationship between the refractive parameters and other important characteristics of the materials.

**Figure 2 materials-15-02153-f002:**
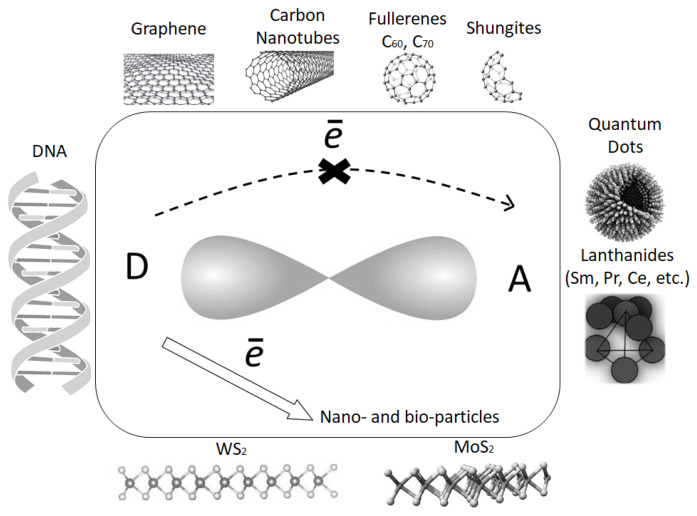
A model showing the formation of the intermolecular CTC.

**Figure 3 materials-15-02153-f003:**
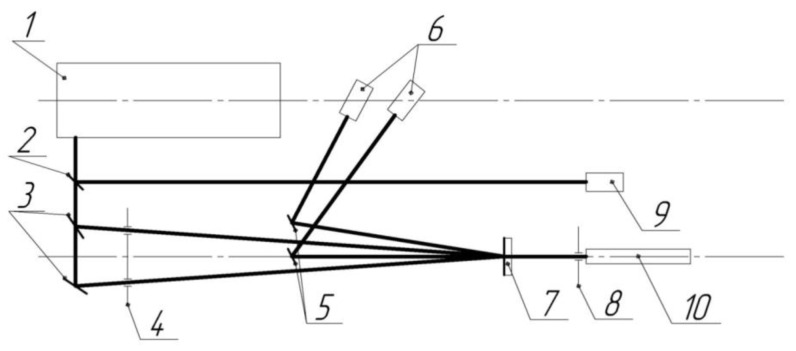
Experimental scheme used to study the materials in the reversible mode.

**Figure 4 materials-15-02153-f004:**
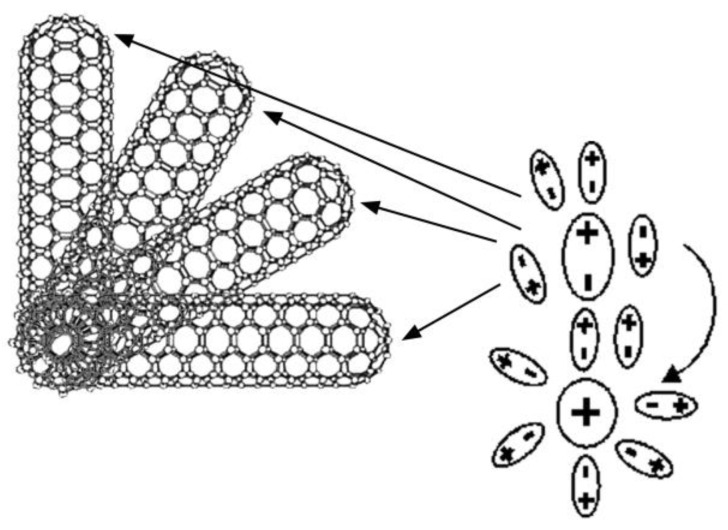
Possible consideration of the orientation of a CNTs end according several dipoles.

**Table 1 materials-15-02153-t001:** Laser-induced change in the refractive index of the studied doped organics.

System	*c*, wt %	λ, nm	*W*_in_, J cm^−2^	Λ, mm^−1^	τ, ns	Δ*n_i_*	References
Pure PI	0	532	0.6	90	20	10^−4^–10^−5^	[39]
PI + malachite green	0.2	532	0.5–0.6	90–100	10–20	2.87 × 10^−4^	[40]
PI + CdSe(ZnS) QDs	0.003	532	0.2–0.3	90–100	10	2.0 × 10^−3^	[41,42]
PI + CdSe(ZnS) QDs	0.03	532	0.2	90–100	10	2.2 × 10^−3^	Current data
PI + shungite	0.1	532	0.6	100	10	3.6 × 10^−3^	Current data
PI + shungite	0.1	532	0.6	150	10	3.46 × 10^−3^	[43]
PI + shungite	0.1	532	0.6	170	10	3.1 × 10^−3^	[43]
PI + shungite	0.2	532	0.063–0.1	150	10	3.8–5.3 ×10^−3^	[44]
PI + shungite	0.2	532	0.5	150	10	4.6 × 10^−3^	Current data
PI + C_60_	0.2	532	0.5–0.6	90	10–20	4.2 × 10^−3^	[39]
PI + C_70_	0.2	532	0.6	90	10–20	4.68 × 10^−3^	[39]
PI + C_70_	0.5	532	0.6	90	10–20	4.87 × 10^−3^	[39]
PI + CNTs	0.05	532	0.3	150	10	4.5 × 10^−3^	[45]
PI + CNTs	0.1	532	0.5–0.8	90	10–20	5.7 × 10^−3^	[39]
PI + CNTs	0.1	532	0.3	150	10	5.5 × 10^−3^	[39,43]
PI + RGO	0.1	532	0.2	100	10	3.4 × 10^−3^	[46]
PI + RGO	0.1	532	0.2	150	10	3.1 × 10^−3^	Current data
PDLC based on PI + C_70_	0.1	532	0.3	100	10	1.15 × 10^−3^	Current data
PDLC based on PI + C_70_	0.2	532	0.3	100	10	1.35 × 10^−3^	Current data
PDLC based on COANP * + C_70_	5	532	17.5 × 10^−3^	90–100	10–20	1.4 × 10^−3^	[39]
PDLC based on COANP + CNTs	0.5	532	18.0 × 10^−3^	90–100	10–20	3.2 × 10^−3^	[39]

* COANP: 2-cyclo-octyl-amine-5-nitropyridine.

## Data Availability

Using these primary materials of the article, as well as taking into account earlier developments on other materials, a patent will be created, as well as a message will be published in the media, for example, at LETI University, since the results are useful for teaching students.

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
