# Peer review of "Advances in Material Nanosensitization: Refractive Property Changes as the Main Parameter to Indicate Organic Material Physical–Chemical Feature Improvements"

_materials, 2022, doi:10.3390/ma15062153_

Round 1

Reviewer 1 Report

The paper seems well constructed and presented albeit with a number of minor issues in the use of English; these do not detract from the meaning and can readily be corrected through some editorial oversight. 

The main concern I have is that the data are not well discussed or interpreted, and I feel there is not enough data. The author states there is a dependence of the change in refractive index wit ha number of factors. That dependence is not discussed and there are often only two data points to support the statement e.g. changes in concentration where there seem only to be two different concentrations studies under the same conditions. It would be better to present more concentrations and present the data graphically to show trends. 

Hence the conclusions finally drawn will have a more solid base in observed results than they do presently.

Author Response

Dear referee!

Thank you for your comments and useful remarks, that can improve the paper for the understanding.

----------------------------------------

Comments and Suggestions for Authors

1). The paper seems well constructed and presented albeit with a number of minor issues in the use of English; these do not detract from the meaning and can readily be corrected through some editorial oversight. 

2). The main concern I have is that the data are not well discussed or interpreted, and I feel there is not enough data. The author states there is a dependence of the change in refractive index wit ha number of factors. That dependence is not discussed and there are often only two data points to support the statement e.g. changes in concentration where there seem only to be two different concentrations studies under the same conditions. It would be better to present more concentrations and present the data graphically to show trends. 

3). Hence the conclusions finally drawn will have a more solid base in observed results than they do presently.

------------------------------------------------------

My answers are here:

1). About English. Thank you for this remark. It is true that Russian researchers currently have little conversational practice, which does not allow using some semantic combinations understandable to an English-speaking audience. Yes, I believe, that Editor Team can correct my English with good advantage. Thank you in advance!

2). Well, I can include some additional data about your consideration. The paragraphs are included in the text body in the Materials and methods part (set-up – in order to understand the process in the reversible mode), some paragraphs are included in the Results and Discussion part. The additional parts are colored with yellow.

3). Nice remarks! I have extended the Conclusion part.

Thank you for the understanding and nice help!

Best Regards,

Natalia Kamanina

=======================================

Natalia V. Kamanina (Prof., Dr.Sci., PhD)

Head of the lab for Photophysics of media with nanoobjects

Vavilov State Optical Institute

Kadetskaya Liniya V.O., dom.5, korpus 2,

St.- Petersburg, 199053, Russia

Professor of the St.-Petersburg Electrotechnical University (“LETI”),

Part-time Leading Researcher at Nuclear Physics Institute (Gatchina)

Job phone: +7 (812) 327-00-95

Fax: +7 (812) 331-75-58 (for N.V.Kamanina)

http://www.photophysics-lab.org/ 

https://publons.com/researcher/1696479/natalia-kamanina/

https://sciprofiles.com/news-feed

http://rusnor.org/network/webinars/10203.htm

http://www.npkgoi.ru/?module=articles&c=profil&b=7

http://www.nanometer.ru/2007/08/09/liquid_crystal_3905.html

http://www.eltech.ru/ru/fakultety/fakultet-elektroniki/sostav-fakulteta/kafedra-kvantovoy-elektroniki-i-optiko-elektronnyh-priborov/sostav-kafedry

=======================================

Reviewer 2 Report

This paper reviewed nanomaterial-based sensitization for changing organic-based material characteristics. Although the research topic is interesting, the overall description in this paper was insufficient to deliver details of this research topic to broad readers. I would like to recommend that this paper should solve some major points and minor points before considering publication in Materials as follows:

Major points

  1. For Figure 2, the author should provide the related references in the figure directly to give the readers clearer information. For example, the author described that synthesis of various nanomaterials such as graphene, carbon nanotubes, WS2, MoS2, and etc. can lead intermolecular charge-transfer-complex (CTC), but there were no specific references so that it is hard to understand in detail.
  2. In the section of Materials and Methods, the author needs to provide an additional representative figure which can explain the measurement setup and material preparation process.
  3. In the section of Results and Discussion, I would like to recommend that the author kindly gives additional figure which can express the additional effects that the author wrote for the broad readers in new application fields.
  4. In short, the author introduced many charming aspect and research results. However, the paper is in lack of explanation to deliver the overall understanding to the broad readers.

Minor points

Some English errors in words and grammar have been detected. I recommend that the authors should conduct proof reading and English revision for any of the errors.

Author Response

Dear referee!

Thank you for your comments and useful remarks, that can improve the paper for the understanding.

-------------------------------------

Comments and Suggestions for Authors

This paper reviewed nanomaterial-based sensitization for changing organic-based material characteristics. Although the research topic is interesting, the overall description in this paper was insufficient to deliver details of this research topic to broad readers. I would like to recommend that this paper should solve some major points and minor points before considering publication in Materials as follows:

Major points

  1. For Figure 2, the author should provide the related references in the figure directly to give the readers clearer information. For example, the author described that synthesis of various nanomaterials such as graphene, carbon nanotubes, WS2, MoS2, and etc. can lead intermolecular charge-transfer-complex (CTC), but there were no specific references so that it is hard to understand in detail.
  2. In the section of Materials and Methods, the author needs to provide an additional representative figure which can explain the measurement setup and material preparation process.
  3. In the section of Results and Discussion, I would like to recommend that the author kindly gives additional figure which can express the additional effects that the author wrote for the broad readers in new application fields.
  4. In short, the author introduced many charming aspect and research results. However, the paper is in lack of explanation to deliver the overall understanding to the broad readers.

Minor points

5). Some English errors in words and grammar have been detected. I recommend that the authors should conduct proof reading and English revision for any of the errors.

------------------------------------
My answers are here:

1). Good remarks! But, sorry, I have included some references in the references list according to fullerenes, reduced graphene oxides, carbon nanotubes, etc. during the materials explanation. Only MoS2-based NPs I have not added in the ref.list. We will collect the data about the influence of the MoS2-based NPs and will publish in the future paper. About the fullerene influence the paper [21-23, 25, 26, 27], lantanoids NPs influence – paper [24], carbon nanotubes and QDs influence – [39-42], shungites – [44,46], etc.

2). Well. I have added the experimental scheme. Please see Figure 3 now.

3). Thank you! I have added in the Results and Discussion part the paragraph about the COANP, NPP, PNP conjugated materials, which have been studied for the optical limiting, hologram recording, display technique, etc.

4). Thank you! I have added some paragraphs in the Results and discussion part.

5). About English. Thank you for this remark. It is true that Russian researchers currently have little conversational practice, which does not allow using some semantic combinations understandable to an English-speaking audience. Yes, I believe, that Editor Team can correct my English with good advantage. Thank you in advance!

I would like to tell that all paragraphs added in this version of the revised paper are collared in yellow.

Thank you for the understanding and nice help!

Best Regards,

Natalia Kamanina

=======================================

Natalia V. Kamanina (Prof., Dr.Sci., PhD)

Head of the lab for Photophysics of media with nanoobjects

Vavilov State Optical Institute

Kadetskaya Liniya V.O., dom.5, korpus 2,

St.- Petersburg, 199053, Russia

Professor of the St.-Petersburg Electrotechnical University (“LETI”),

Part-time Leading Researcher at Nuclear Physics Institute (Gatchina)

Job phone: +7 (812) 327-00-95

Fax: +7 (812) 331-75-58 (for N.V.Kamanina)

http://www.photophysics-lab.org/ 

https://publons.com/researcher/1696479/natalia-kamanina/

https://sciprofiles.com/news-feed

http://rusnor.org/network/webinars/10203.htm

http://www.npkgoi.ru/?module=articles&c=profil&b=7

http://www.nanometer.ru/2007/08/09/liquid_crystal_3905.html

http://www.eltech.ru/ru/fakultety/fakultet-elektroniki/sostav-fakulteta/kafedra-kvantovoy-elektroniki-i-optiko-elektronnyh-priborov/sostav-kafedry

=======================================

Reviewer 3 Report

It is very difficult to understand the content of the article due to several reasons. First, the English style and grammar used makes it difficult to understand. Secondly, the author refers to the methods and results obtained in different articles, but she does not make any introduction, description or present any graph or scheme, for example of the pattern of the samples after laser irradiation, the measurement of the different magnitudes, the models used. Finally, it has been shown that laser irradiation can produce different effects such as oxidation or excision of the polymer chains, which most probably also change the refractive index of the material. Has the author studied these effects?

Author Response

Dear referee!

Thank you for your comments and useful remarks, that can improve the paper for the understanding.

-----------------------------------

Comments and Suggestions for Authors

It is very difficult to understand the content of the article due to several reasons. First, the English style and grammar used makes it difficult to understand. Secondly, the author refers to the methods and results obtained in different articles, but she does not make any introduction, description or present any graph or scheme, for example of the pattern of the samples after laser irradiation, the measurement of the different magnitudes, the models used. Finally, it has been shown that laser irradiation can produce different effects such as oxidation or excision of the polymer chains, which most probably also change the refractive index of the material. Has the author studied these effects?

-------------------------------------------

My answers are here:

Thanks! Excellent reasoning. My vision of the problem of working with polyimide is as follows. This unique material, apparently the best of polymer compositions, has a melting point of more than 900-1000 degrees. In my laboratory, we study these materials poured onto the substrates both recently and 5-10-15 years ago. When structuring with carbon nanoparticles, the material becomes more durable and does not lose its properties after exposure to laser radiation at the wavelength of 532 nm.

As for the pure polyimide composition. Earlier, the destruction limit was found (during the transition from the reversible to the irreversible recording mode of the amplitude-phase hologram. This limit is 0.5-0.6 J cm-2 for 532 nm irradiation (Please see the ref.57 in the ref.list). Quite a large data for polymer materials. I have put the reference about this threshold in the reference list. It is colored with yellow.

Recently we have started the irradiation of the polyimide in the vacuum UV range. In this range we found some destruction of the polyimide materials at the wave length of 126 nm  and at 172 nm due to some due to the ablation ((Please see the ref.58 in the ref.list). We collect now the data about the irradiation of the Polyimide, COANP, NPP, PNP, etc. materials and I will make the paper in future including my team as the co-authors as well. The rules for the Russian scientists are the follow: to publish some data in the Russian journal, after that we have the chance to publish in the international journals.

I should draw your kind attention that all parts, which I have added in the text body, are collared in yellow.

Thank you for the understanding and nice help!

Best Regards,

Natalia Kamanina

=======================================

Natalia V. Kamanina (Prof., Dr.Sci., PhD)

Head of the lab for Photophysics of media with nanoobjects

Vavilov State Optical Institute

Kadetskaya Liniya V.O., dom.5, korpus 2,

St.- Petersburg, 199053, Russia

Professor of the St.-Petersburg Electrotechnical University (“LETI”),

Part-time Leading Researcher at Nuclear Physics Institute (Gatchina)

Job phone: +7 (812) 327-00-95

Fax: +7 (812) 331-75-58 (for N.V.Kamanina)

http://www.photophysics-lab.org/ 

https://publons.com/researcher/1696479/natalia-kamanina/

https://sciprofiles.com/news-feed

http://rusnor.org/network/webinars/10203.htm

http://www.npkgoi.ru/?module=articles&c=profil&b=7

http://www.nanometer.ru/2007/08/09/liquid_crystal_3905.html

http://www.eltech.ru/ru/fakultety/fakultet-elektroniki/sostav-fakulteta/kafedra-kvantovoy-elektroniki-i-optiko-elektronnyh-priborov/sostav-kafedry

=======================================

Round 2

Reviewer 2 Report

After careful concern, I regret that I am not able to recommend this paper to be published in Materials.

The main points for the reason are as follows:

  1. Although I requested that the author should make the paper more readable and understandable in the revised manuscript, it is still not sufficient. For instance, the author tried to review the previous literatures by using only writing, which makes readers unattractive to read this review. The author should bring some figures from the previous literatures to give more detailed information to the readers.
  2. Thus, this paper has been written by author-centered explanation, so it is unable to appeal and make an impact to broader readers in material research. In particular, it is too hard to read and understand this paper because the author made too short comments for each point of the research.

In short, the author should make the manuscript more visualized in order to give attractive review to broad readers.

Author Response

Dear referee!

Thanks a lot for your recommendation. I have seen once again the paper and try to improve some paragraphs in order to more good for the readers. You can see the revised file.

-----------------------------------------------------------

Comments and Suggestions for Authors

After careful concern, I regret that I am not able to recommend this paper to be published in Materials.

The main points for the reason are as follows:

  1. Although I requested that the author should make the paper more readable and understandable in the revised manuscript, it is still not sufficient. For instance, the author tried to review the previous literatures by using only writing, which makes readers unattractive to read this review. The author should bring some figures from the previous literatures to give more detailed information to the readers.
  2. Thus, this paper has been written by author-centered explanation, so it is unable to appeal and make an impact to broader readers in material research. In particular, it is too hard to read and understand this paper because the author made too short comments for each point of the research.

In short, the author should make the manuscript more visualized in order to give attractive review to broad readers.

Forgive me, but I don't quite understand your claims to the article. Yes, this is my local opinion about nonlinear optical processes in matter. The model representation is substantially supported by analytical calculations and a large series of experiments. The model has been tested at many scientific and technical sites in Europe, America, and Russia. It is really we can testify that the refractive index is more important parameters for the classical and essentially for the doped organics. I think that you are not quite right in your claims to my article. I'm sorry again.

All paragraphs improved have been collared with green.

Thank you very much once again for your job regarding to paper improving.

Best Regards,

Natalia Kamanina

=======================================

Natalia V. Kamanina (Prof., Dr.Sci., PhD)

Head of the lab for Photophysics of media with nanoobjects

Vavilov State Optical Institute

Kadetskaya Liniya V.O., dom.5, korpus 2,

St.- Petersburg, 199053, Russia

Professor of the St.-Petersburg Electrotechnical University (“LETI”),

Part-time Leading Researcher at Nuclear Physics Institute (Gatchina)

Job phone: +7 (812) 327-00-95

Fax: +7 (812) 331-75-58 (for N.V.Kamanina)

http://www.photophysics-lab.org/ 

https://publons.com/researcher/1696479/natalia-kamanina/

https://sciprofiles.com/news-feed

http://rusnor.org/network/webinars/10203.htm

http://www.npkgoi.ru/?module=articles&c=profil&b=7

http://www.nanometer.ru/2007/08/09/liquid_crystal_3905.html

http://www.eltech.ru/ru/fakultety/fakultet-elektroniki/sostav-fakulteta/kafedra-kvantovoy-elektroniki-i-optiko-elektronnyh-priborov/sostav-kafedry

=======================================

Reviewer 3 Report

The author has responded to some of the issues raised in the first review, however, it is necessary to improve the English so that the reader can follow the argumentation of the article.

Author Response

Dear referee!

Thanks a lot for your recommendation. I have seen once again the paper and try to improve English. You can see the revised file.

--------------------------------------------------

Comments and Suggestions for Authors

The author has responded to some of the issues raised in the first review, however, it is necessary to improve the English so that the reader can follow the argumentation of the article.

-----------------------------------------------

All paragraphs improved have been collared with green.

Thank you very much once again for your job regarding to paper improving.

Best Regards and be healthy!

Natalia Kamanina

=======================================

Natalia V. Kamanina (Prof., Dr.Sci., PhD)

Head of the lab for Photophysics of media with nanoobjects

Vavilov State Optical Institute

Kadetskaya Liniya V.O., dom.5, korpus 2,

St.- Petersburg, 199053, Russia

Professor of the St.-Petersburg Electrotechnical University (“LETI”),

Part-time Leading Researcher at Nuclear Physics Institute (Gatchina)

Job phone: +7 (812) 327-00-95

Fax: +7 (812) 331-75-58 (for N.V.Kamanina)

http://www.photophysics-lab.org/ 

https://publons.com/researcher/1696479/natalia-kamanina/

https://sciprofiles.com/news-feed

http://rusnor.org/network/webinars/10203.htm

http://www.npkgoi.ru/?module=articles&c=profil&b=7

http://www.nanometer.ru/2007/08/09/liquid_crystal_3905.html

http://www.eltech.ru/ru/fakultety/fakultet-elektroniki/sostav-fakulteta/kafedra-kvantovoy-elektroniki-i-optiko-elektronnyh-priborov/sostav-kafedry

=======================================
